

# Overexpressed PLAU and its potential prognostic value in head and neck squamous cell carcinoma

Zhexuan Li[1,2,3], Changhan Chen[1,2,3], Juncheng Wang[1,2,3], Ming Wei[1,2,3], Guancheng Liu[1,2,3], Yuexiang Qin[1,2,3], Li She[1,2,3], Yong Liu[1,2,3,4], Donghai Huang[1,2,3,4], Yongquan Tian[1,2,3], Gangcai Zhu[5] and Xin Zhang[1,2,3,4]

[1] Department of Otolaryngology-Head and Neck Surgery, The Xiangya Hospital, Central South University, Changsha, Hunan, China
[2] Otolaryngology Major Disease Research Key Laboratory of Hunan Province, Changsha, Hunan, China
[3] Clinical Research Center for Pharyngolaryngeal Diseases and Voice Disorders in Hunan Province, Changsha, Hunan, China
[4] National Clinical Research Center for Geriatric Disorders, Changsha, Hunan, China
[5] Department of Otolaryngology-Head and Neck Surgery, The Second Xiangya Hospital, Central South University, Changsha, Hunan, China

Corresponding author
Gangcai Zhu,
qianhudoctor@csu.edu.cn

## ABSTRACT

**Background**. Metastasis is a major event for survival and prognosis in patients with head and neck squamous cell carcinomas (HNSCC). A primary cause of metastasis is the proteolytic degradation of the extracellular matrix (ECM). The plasminogen activator urokinase (PLAU) is involved in the transformation of plasminogen to plasmin leading to hydrolyzation of ECM-related proteins. However, the role of PLAU expression in HNSCC is unclear and the worth being investigated.

**Methods**. PLAU expression profiles and clinical parameters from multiple HNSCC datasets were used to investigate the relationship of PLAU expression and HNSCC survival. GO and PPI network were established on PLAU-related downstream molecular. The stroma score was deconvoluted for analysis of PLAU's association with the immune environment. ROC analysis was applied to show the performance of PLAU in predicting HNSCC prognosis.

**Results**. PLAU mRNA was significantly elevated, as opposed to its methylation, in HNSCC tumor samples over normal specimens (all $p < 0.01$). Univariate and multivariate cox analysis showed PLAU could be an independent indicator for HNSCC prognosis. Combining with neck lymph node status, the AUC of PLAU in predicting 5-years overall survival reached to 0.862. GO enrichment analysis showed the major biological process (extracellular matrix organization and the P13K-Akt signaling pathway) may involve to the possible mechanism of PLAU's function on HNSCC prognosis. Furthermore, PLAU expression was positively correlated with stroma cell score, M1 type macrophages, and negatively associated with CD4 + T cell, Tregs cell, and follicular helper T cell.

**Conclusions**. PLAU might be an independent biomarker for predicting outcomes of HNSCC patients. The elevated expression of PLAU was associated with HPV positivity and neck node status. The PI3K-Akt pathway and aberrant proportions of immune cells might underly the mechanism of PLAU's oncogene role in HNSCC.

## INTRODUCTION

Head and neck squamous cell carcinomas (HNSCC) are among the most aggressive malignancies and over 50% of patients present with locally advanced or metastatic disease (*Torre et al., 2015*). More than 830,000 patients are diagnosed and over 430,000 patients die from this disease worldwide annually (*Cramer et al., 2019*). This disease is characterized by low survival rates, high recurrence rates, and/or regional lymph node that become metastatic (*Siu et al., 2019*). Although the examination and treatment has been improving in recent decades, the overall 5-year survival rate of HNSCC patients does not increase remarkably (*Yang et al., 2019*). Prognosis prediction is crucial for physicians to offer consultants and personized treatment. However, clinical parameters such as TNM classification are the main sources physicians generally relying on for predicting patient outcome and making therapeutic decision, which is inaccurate in many situation (*Kowalski et al., 2005*; *Moertel et al., 1995*). It is well accepted that molecular biomarkers may facilitate the prognosis prediction for SCCHN patients (*Kang, Kiess & Chung, 2015*; *Leemans, Snijders & Brakenhoff, 2018*). Currently, there is no matured biomarkers is approved for HNSCC prognosis prediction. Therefore, it is expected and worth that the identification of novel biomarkers assisting with patient care and survival improvement.

Metastasis is one of the major events leading to unfavorable survival time for HNSCC patients (*Chen et al., 2018*). The mechanism of HNSCC metastasis is unknown, accumulated evidence show that ECM reconstruction may involve providing a physical and biochemical niche for humor cell metastasis (*Hanahan & Weinberg, 2011*; *Murphy & Courtneidge, 2011*).

PLAU belonging to the S1 serine peptidase of Clan PA, also named Urokinase-type plasminogen activator (uPA) is a proteinase involving in the transformation of plasminogen to plasmin (*Ai et al., 2020*), and it could hydrolyze ECM remodeling related proteins and activates growth factors (*Danø et al., 2005*). Some studies report that the expression level of PLAU is significantly correlated to tumor cell lymph node and distant organ metastasis (*Gutierrez et al., 2000*). Emerging evidence implies that PLAU plays a critical role in the initiation and development of various cancers including breast cancer, colorectal cancer, and esophageal cancer (*Li et al., 2017*; *Lin et al., 2019*; *Novak et al., 2019*). However, the role of PLAU needs to be explored further in HNSCC. Here, we applied multiple datasets to evaluate the increased expression of PLAU in HNSCC tumor samples as compared to adjacent tissues and confirm it as an independent prognosis predictor of HNSCC patients in different angles and levels. The co-expression network and scores of tumor immune microenvironment were established and analyzed in this study as well, which could be interpreted to the possible mechanism of PLAU's role in HNSCC patients.

**Table 1  Cutoff identification for survival time by clinical parameters.**

| Index | Cutoff | AUC | 95% CI | Sensitivity | Specificity | PPV | NPV |
|-------|--------|-----|--------|-------------|-------------|-----|-----|
| T | / | 0.532 | 0.451∼0.616 | 0.711 | 0.474 | 0.523 | 0.669 |
| N | / | 0.58 | 0.495∼0.668 | 0.768 | 0.437 | 0.549 | 0.679 |
| HPV | / | 0.531 | 0.437∼0.624 | 0.702 | 0.458 | 0.569 | 0.602 |
| PLAU | 9532 | 0.795 | 0.724∼0.863 | 0.901 | 0.679 | 0.749 | 0.798 |
| N+PLAU | / | 0.862 | 0.663∼0.887 | 0.885 | 0.639 | 0.704 | 0.801 |
| T+PLAU | / | 0.778 | 0.821∼0.906 | 0.995 | 0.619 | 0.748 | 0.812 |

**Notes.**

PPV, Positive Predictive Value; NPV, Negative Predictive Value.

# MATERIALS & METHODS

## Data collection and normalization

GSE25099 from 79 HNSCC patients (*Peng et al., 2011*), GSE13601 consisting of 37 HNSCC patients (*Estilo et al., 2009*), GSE65858 from 270 HNSCC patients (*Wichmann et al., 2015*), GSE136037 from 49 HNSCC patients (*Alfieri et al., 2020*), and The Cancer Genome Atlas (TCGA) -HNSC cohorts including 546 HNSCC HTSeq-counts, methylation profiles and related clinical information were downloaded. CalcNormFactors was used to calculate normalization factors to scale the gene expression in TCGA dataset (*Anders & Huber, 2010*; *Le et al., 2019*). Youden index (sensitivity + specificity -1) was used to calculate the best cutoff of survival analysis by R package ("SurvivalROC") (*Huang, Liao & Li, 2017*; *Luo & Xiong, 2013*). The GSE65858 and GSE136037 datasets from the GEO database were converted to transcripts per million (TPM) (*Zhao, Ye & Stanton, 2020a*). And the term "N-" means primary lesions in HNSCC patients without neck lymph node metastasis, and "N+" means primary lesions in HNSCC patients with neck lymph node metastasis. The normalized data provided from original studies in other datasets were used in this study directly.

## Survival analysis

Briefly, the expression of PLAU was categorized into low or high by 'Maximally Selected Rank Statistics' (maxstat) method (*Lausen & Schumacher, 1992*). The cutoff value of PLAU mRNA expression in TCGA was 9,532 in Table 1. Older or younger is classified based on its mean age. The overall survival (OS), progression-free interval (PFI) or recurrence free survival (RSF) curves were visualized by Kaplan–Meier plots. Univariate and multivariate Cox regression analysis was applied to death hazard ratios calculation after the proportional hazard assumption was tested.

## Go analysis and co-expression network establishment

Go enrichment analysis was performed by Bioconductor package "clusterProfiler" (*Yu et al., 2012*). The method of KEGG enrichment analysis was performed same as the Go enrichment analysis. The co-expression genes were screened using R packages ("limma").From the TCGA-HNSCC databases, we used Pearson correlation coefficients (|Pearson correlation coefficient| > 0.5 and $P$-value <0.001) and the $z$-test to examine the correlation between PLAU expression level and co-expression genes. STRING database

and Cytoscape tool were used to construct the protein-protein-interactions (PPI) of genes (*Struk & Jacobs, 2019*).

## Immune cell environment analysis

Estimation of stromal and immune cells in malignant tumor tissues using Expression data (ESTIMATE) is a tool for predicting tumor purity, and the presence of infiltrating stromal/immune cells in tumor tissues using gene expression data (*Yoshihara et al., 2013*). We used R packages ("estimate") to get immune-environment scores in HNSCC patients. R packages ("CIBERSORT.R", https://cibersortx.stanford.edu/) (*Newman et al., 2015*) was used to deconvoluted 22 common immune cell proportions in HNSCC patients. The correlation of PLAU expression with immune environment and 22 immune cells were investigated by Pearson test.

## Software and statistical analyses

GraphPad Prism 8 or R studio (version 3.5.3) was used to evaluate all data (*Kamdem et al., 2019*; *Le & Huynh, 2019*). Chi-squre or Fisher's exact test was performed to compare the differences expression of PLAU across different groups. $P < 0.05$ was considered statistically significant. The detailed codes and other packages' information were included in the supplementary materials.

# RESULTS

## PLAU mRNA is over-expressed in HNSCC

To discover the expression of PLAU mRNA in HNSCC, we analyzed three independent patient cohorts, which showed a consistent result that PLAU mRNA expression was elevated in HNSCC tumors than normal tissues(all $p < 0.01$, Figs. 1A–1C). And the overexpression of PLAU mRNA was confirmed in 10 different HNSCC cell lines as compared to 4 types of human keratinocyte cell lines ($p = 0.002$, Fig. 1D). Furthermore, PLAU mRNA in HPV positive HNSCC samples was interposed between adjacent normal tissues and HPV negative tumors (Fig. S1A).

There was significantly less PLAU mRNA in HPV positive tumors than HPV negative ones according to another patient cohort (Fig. S1B).

## Association of PLAU mRNA with neck lymph node status in HNSCC

In order to study the role of PLAU mRNA may play in HNSCC patients, the relationship of PLAU mRNA and clinical parameters was further characterized in TCGA-HNSC cohort. As shown in Table 2, there was no significantly different expression of PLAU mRNA in different age, gender, clinical stage, and tumor stage, but the difference of PLAU in HPV positivity and neck node status was considered as significant ($p = 0.001$, $p = 0.033$, respectively). Higher expression of PLAU was founded in patients with neck lymph node metastasis than patient without neck lymph node metastasis in another independent cohort as well (Fig. 1E).

## PLAU is an independent predictor of HNSCC prognosis

Considering the above findings, we continued to analysis the possible correlation of HNSCC survival time and PLAU expression. As shown in Table 3, age, clinical stage, tumor size, neck

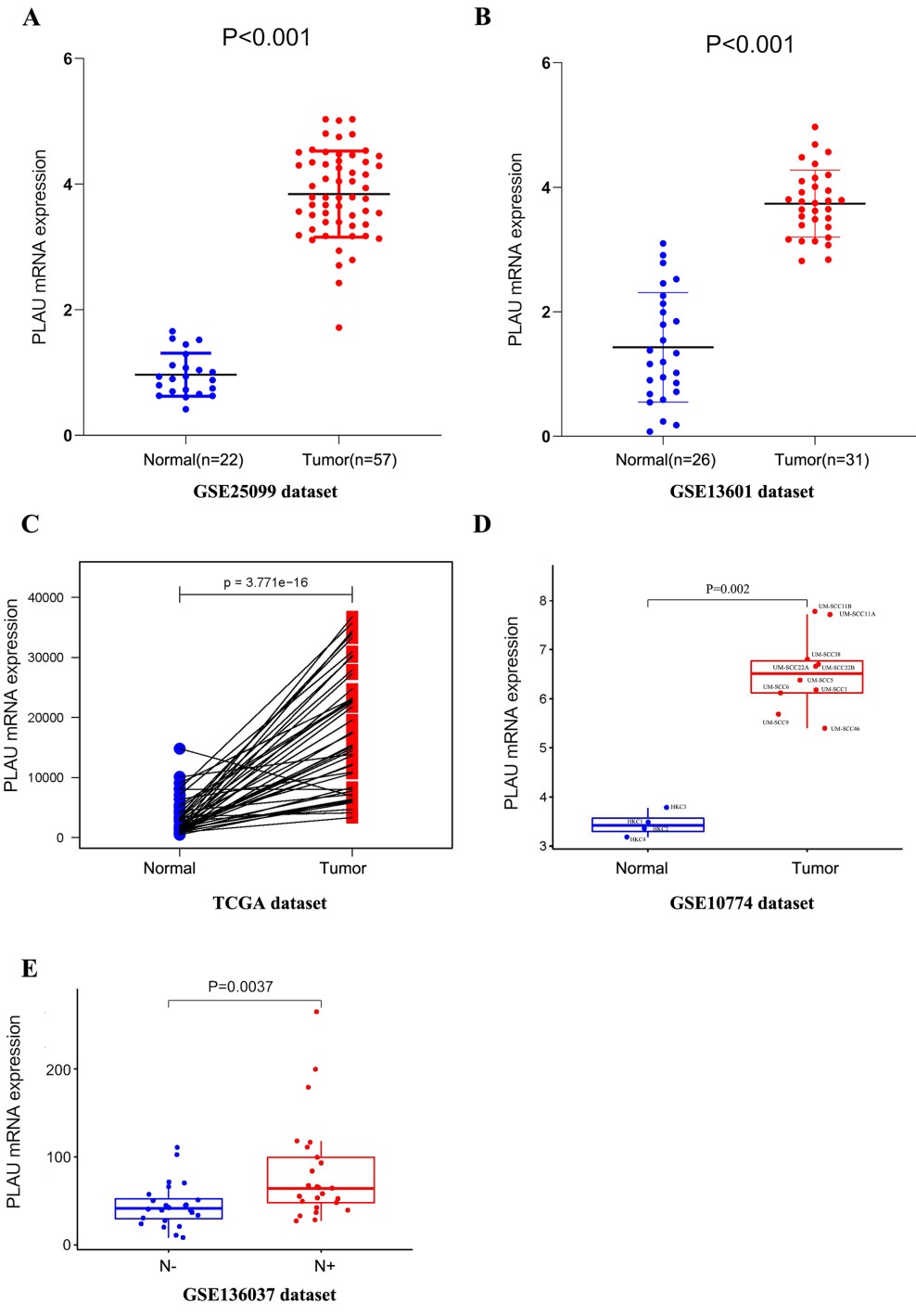

**Figure 1** **PLAU mRNA is over-expressed in HNSCC.** (A, B) The expression of PLAU mRNA in normal and HNSCC tumor tissues was detected from the GEO databases. (C) Compared the difference expression of PLAU mRNA between tumor tissues and pair normal tissues in HNSCC. (D) The overexpression of PLAU mRNA was confirmed in HNSCC cell lines and human keratinocyte cell lines. (E) The difference expression of PLAU mRNA in neck lymph node status. "N-" indicates patients without neck lymph node metastasis, and "N+" indicates patients with neck lymph node metastasis.

**Table 2  The PLAU expression in HNSCC patients with different clinical parameters.**

| Clinical parameters | PLAU mRNA expression | | P-value |
|---|---|---|---|
| | Low (n = 184) | High (n = 312) | |
| **Age(years)** | | | 0.949 |
|    Mean (SD) | 61.6 (11.8) | 60.7 (12.0) | |
|    Median [Min, Max] | 61.0 [26.0, 87.0] | 60.5 [19, 90] | |
| **Gender** | | | 0.943 |
|    Female | 49(36.8%) | 84(37.2%) | |
|    Male | 135(63.2%) | 228(62.8%) | |
| **Clinical stage** | | | 0.913 |
|    I-II | 136(37%) | 232(37.5%) | |
|    III-IV | 48(63%) | 80(62.5%) | |
| **Tumor stage** | | | 0.23 |
|    T1-2 | 88(47.8%) | 126(40.4%) | |
|    T3-4 | 93(50.6%) | 182(58.3%) | |
|    Missing | 3(1.6%) | 4(1.3%) | |
| **Neck nodal metastasis** | | | 0.033* |
|    N- | 82(44.6%) | 105(33.7%) | |
|    N+ | 101(54.9%) | 202(64.7%) | |
|    Missing | 1(0.5%) | 5(1.6%) | |
| **HPV** | | | 0.001*** |
|    HPV- | 134(72.8%) | 270(86.5%) | |
|    HPV+ | 49(26.7%) | 38(12.2%) | |
|    Missing | 1(0.5%) | 4(1.3%) | |

Notes.
  *$P < 0.05$.
  **$P < 0.01$.
  ***$P < 0.005$.

lymph node status, HPV positivity and PLAU expression was considered to be significantly associated with overall survival time in univariate cox analysis of 496 HNSCC patients. Multivariate cox analysis indicated the hazard ratio of death was reached to 1.52 when high PLAU expressed HNSCC patients compared to patients with low PLAU expression after excluding the potential affections from age, tumor size, neck node metastasis and HPV positivity (Fig. 2A, $p = 0.012$, 95% CI [1.09–2.10]).

The survival curves of PLAU expression were visualized by Kaplan–Meier plots (Fig. 2B), which implied that high PLAU expressed HNSCC patients had decreased overall survival probability than patients with low PLAU expression (Fig. 2B). The same finding could be observed in another independent HNSCC cohort (Fig. 2C). Additionally, high expression of PLAU in HNSCC patients was founded to predict unfavorable outcomes in terms of PFI and RSF (Figs. 2D–2E).

**Table 3   The hazard ratio of PLAU expression and clinical parameters in 496 HNSCC patients.**

| Parameter | Univariate analysis | | |
|---|---|---|---|
| | HR | 95% CI | *P*-value |
| **Age** | | | 0.018[*] |
| Older vs. Younger | 1.021 | 1.003~1.034 | |
| **Gender** | | | 0.087 |
| Female vs. Male | 0.728 | 0.507~1.047 | |
| **Clinical stage** | | | <0.001[***] |
| III–IV vs I–II | 2.902 | 1.605~5.25 | |
| **Tumor size** | | | 0.003[**] |
| T3+T4 vs. T1+T2 | 1.562 | 1.158~2.109 | |
| **Neck node metastasis** | | | <0.001[***] |
| N+ vs. N- | 1.725173 | 1.258~2.365 | |
| **HPV** | | | <0.001[***] |
| HPV+ vs. HPV- | 0.468349 | 0.312~0.703 | |
| **PLAU** | | | 0.001[***] |
| High vs. Low | 1.667282 | 1.218–2.282 | |

Notes.
[*]$P < 0.05$.
[**]$P < 0.01$.
[***]$P < 0.001$.

## Performance of PLAU expression in predicting 5-year overall survival outcomes of HNSCC patients

The predictive performance of PLAU expression in 5-year overall survival outcomes of HNSCC patients was analyzed by ROC analysis. As shown in Fig. 3, the areas under the ROC curve (AUCs) of PLAU expression was 0.795, higher than HPV status (AUC: 0.531), neck node status (AUC: 0.58) and tumor size stage (AUC: 0.532). Combining PLAU expression and neck node status as an integrated factor (N+PLAU), the AUC was 0.862, with an 88.5% sensitivity, 64% specificity, 70.4% positive predictive value and 80.1% negative predictive value. Taken together, PLAU itself or combined with neck node status may facilitate the overall survival prediction for HNSCC patients.

## Hypomethylation of PLAU in HNSCC patients

A growing body of literature reports that DNA methylation and mRNA expression most likely to be negative correlated (*Huang et al., 2015*). Methylation profiles of 496 TCGA-HNSC patients were analyzed to investigate the possible upstream reasons for elevated PLAU mRNA expression in HNSCC tumors. As shown in Fig. 4, the methylated level of PLAU was successively and significantly decreased in adjacent normal tissues, HPV positive tumors and HPV negative tumors (Fig. 4A). The PLAU methylation and mRNA expression was negatively correlated (Fig. 4B, $R = -0.43$, $p < 0.001$). As expected, patients with hypomethylated PLAU, indicating higher expression of PLAU mRNA, had worse overall survival outcomes than patients with hypermethylated PLAU (Fig. 4C).

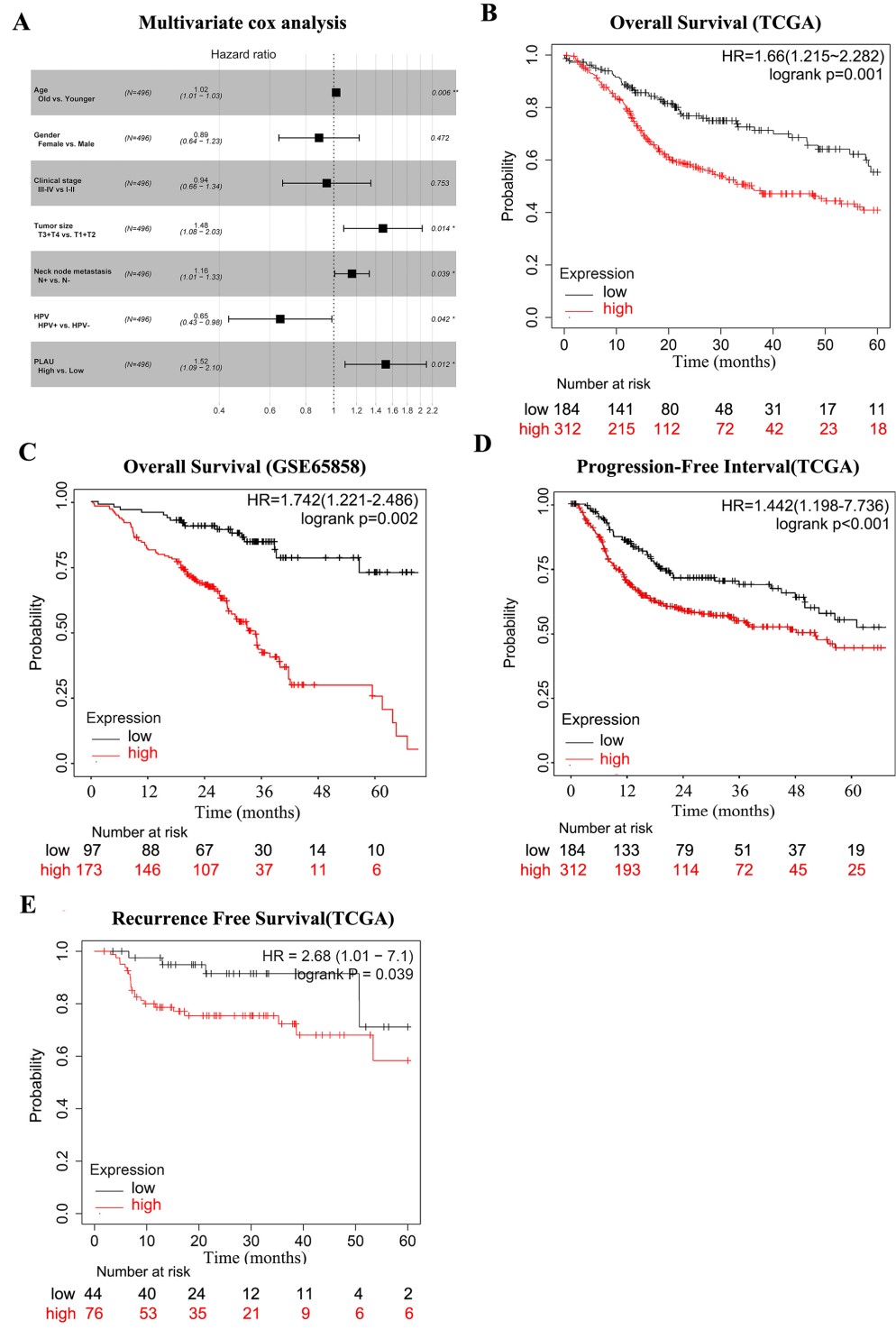

**Figure 2** **PLAU is an independent predictor of HNSCC prognosis.** (A) Multivariate cox analysis related to PLAU. (B, C) The TCGA dataset and the GSE65858 database were used to assess the effect of PLAU expression on overall survival (OS). (D,E) The TCGA dataset assessed the effect of PLAU expression on progression-free interval (PFI) and relapse free survival (RFS).

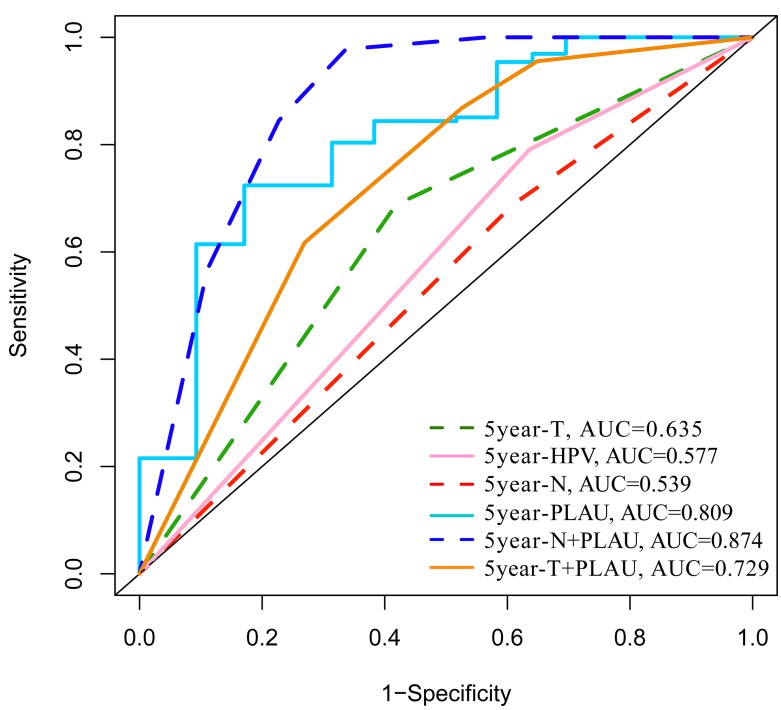

**Figure 3** ROC analysis of the PLAU expression with the clinical parameters of HNSCC in TCGA datasets.

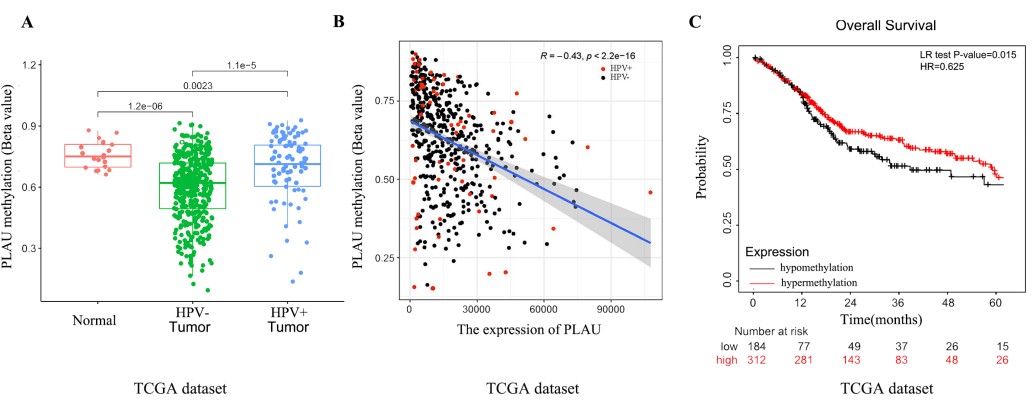

**Figure 4** Relationship between PLAU methylation and HNSCC patients. (A) Separating the groups by HPV status, PLAU methylation compared with the normal samples were evaluated in HNSCC samples of TCGA databases. (B) The correlation between the expression of PLAU and the beta value of methylation were evaluated by TCGA datasets. (C) The methylation of PLAU on OS by TCGA datasets.

## Network establishment for PLAU correlated genes in HNSCC

To further understand the possible downstream reasons a total of 205 genes was correlated with PLAU expression in TCGA-HNSCC patients (21 of 205 genes were negatively correlated with PLAU and 184 genes were positively correlated with PLAU). The top 20 genes of positively or negatively correlated with PLAU are shown in a Heatmap

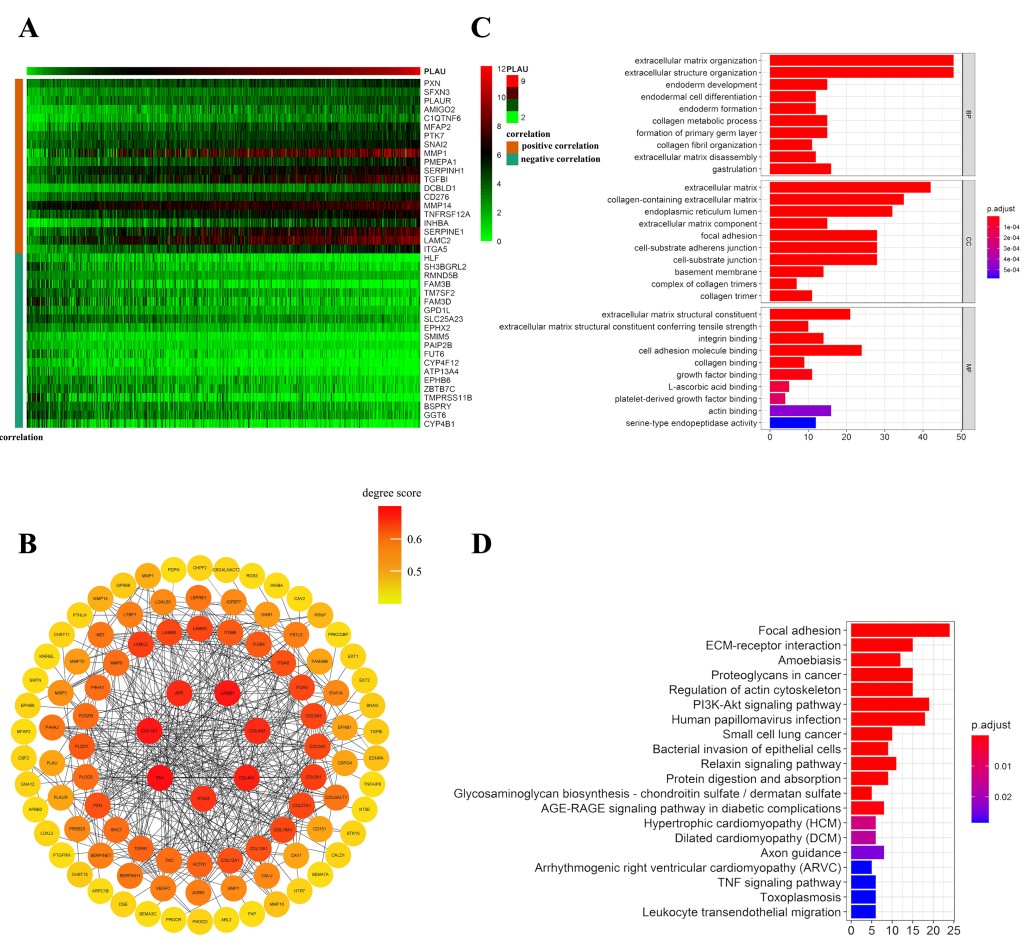

**Figure 5** **Network establishment for PLAU correlated genes in HNSCC.** (A) The top 20 genes of positively or negatively correlated with PLAU were showed in Heatmap. (B) The PPI networks of PLAU interaction partners generated by STRING and Cytoscape. The color represents the degree score (represent the intensity of the hub interacting with its neighbors). Degree score < 0.5 represented low value (colored yellow ), degree score ≥ 0.5 represented high value (colored orange or red). (C) Major biological process, cellular component and molecular functions of PLAU biology by GO enrichment analysis. (D) KEGG pathway analyses further illuminate enriched function pathway related to PLAU.

(Fig. 5A). The interaction network of these 205 genes was established based on STRING and Cytoscape (Fig. 5B). Next, we performed GO and KEGG enrichment analysis to understand the potential biological functions of PLAU in HNSCC. GO analysis showed that the major biological process (extracellular matrix organization), cellular component (extracellular matrix), and molecular functions (cell adhesion molecule binding) may contribute to PLAU related biology (Fig. 5C). KEGG pathway analysis illuminated that the P13K-Akt signaling pathway, human papillomavirus infection, proteoglycans in cancer, and focal adhesion as significantly enriched by the PLAU co-expressed genes (Fig. 5D).

### Distributions of tumor infiltrating immune cell in HNSCC patients with different PLAU expression

More and more evidence revealed the tumor immune microenvironment is a crucial factor in tumor biology (*Mao et al., 2020*). To interpret the role of PLAU expression in HNSCC based on immunity conception, the scores or proportions of tumor infiltrating cells were compared in TCGA-HNSCC cohort. It shows that PLAU expression was positively correlated with stromal score (Fig. 6A). Further analysis found the expression level of PLAU was positively correlated with M1 type macrophages, negatively associated with CD4 + T cell, Tregs cell, and follicular helper T cell (Figs. 6B–6E) (All $p < 0.05$).

## DISCUSSION

Our study, using multiple publicly available profiles in HNSCC cohorts and cell lines, confirmed that PLAU mRNA was over-expressed and associated with neck node lymph metastasis in HNSCC tumors. And, we showed that PLAU expression might be an independent prognosis index for HNSCC patients, which consistent with many other cancer reports (*Mahmood, Mihalcioiu & Rabbani, 2018*) including breast cancer, prostate cancer, ovarian cancer, sarcoma, melanoma, gastric cancer, esophageal cancer, and colorectal cancer. Furthermore, the DNA methylation level and mRNA expression level of PLAU was investigated in our work. As far as we know, this is the first report to link PLAU methylation level with its mRNA expression in cancer samples. There are many possible explanations, such as generic regulation, epigenetic modulation, or mRNA decay, for aberrant expression of mRNA like PLAU. Our work implied that the increased expression of PLAU in HNSCC tumors might be contributed by its hypomethylated levels to some extent. Genetic mutations and epigenetic alterations have critical functions in modulating oncogenes' transcription in human carcinomas (*Wang et al., 2019b*; *Zhou et al., 2019*). The methylation values of DNA could be a prognosis biomarker in cancer (*Teixeira et al., 2019*), which supported our finding that HNSCC patient with hypomethylated PLAU might have a worse survival outcome.

PLAU is a gene encodes for urokinase plasminogen activator (uPA). The detailed mechanism that underlying PLAU's role in HNSCC remain unclear. But it was indicated to involve in the transformation of inactive plasminogen into active plasminogen, which plays an important role in a series of transfer cascades (*Choong & Nadesapillai, 2003*). Previous research has shown that PLAU and type plasminogen activator (tPA) mediated the plasminogen activator (PA) system (*Mahmood, Mihalcioiu & Rabbani, 2018*). PLAU can increase cell proliferation through the activate growth factors or adhesion molecules, for example, VEGF, TGF-β and the α5β1 integrins (*Aguirre-Ghiso et al., 2001*; *Duffy, 2004*; *Ulisse et al., 2009*). PLAU could increase cell adhesion and migration during metastasis and proliferation of tumor cells (*Zhao et al., 2020b*), which may explain our finding that elevated expression of PLAU in node metastasis tumors. Under hypoxia conditions, PLAU expression can activate downstream Akt and Rac1 signaling pathways, thus promoting EMT and cell invasion (*Lester et al., 2007*). In our GO and pathway enrichment assays, PI3K-Akt pathway was enriched by PLAU co-expressed genes. Akt activation maybe the downstream pathway of PLAU leading to HNSCC cell invasion and metastasis.

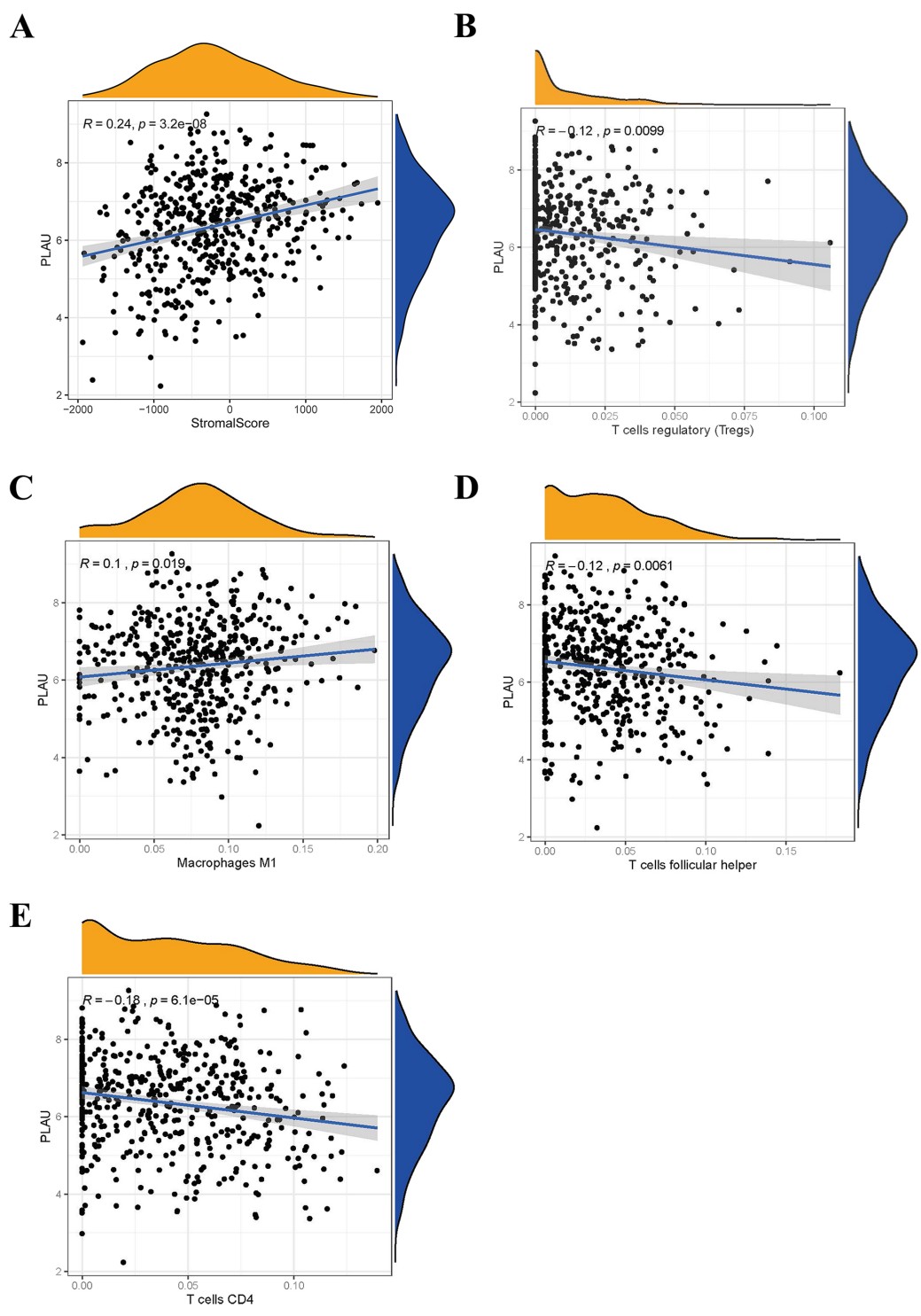

**Figure 6** **Relationship between PLAU and tumor immune microenvironment of HNSCC.** (A) The expression of PLAU is positively correlated with stromal score. (B–E) The expression of PLAU was correlated with Tregs cell, M1 type macrophages, follicular helper T cell and CD4 + T cell.

Growing evidence suggests that cells (such as macrophages, T cells, neutrophils, lymphoid cells and so on) in the immune microenvironment are related to tumor escape and progression (*Hinshaw & Shevde, 2019*). Kipp Weiskopf et al. found that CD47 engaged signal-regulatory protein alpha , which acts as an inhibitory receptor on macrophages to promote immune evasion (*Weiskopf et al., 2016*). Recent data suggest that exposure to immune checkpoint inhibitors (ICI) increase tumor sensitivity to chemotherapy in HNSCC (*Saleh et al., 2019*). Therefore, the investigation of the relationship between HNSCC and the immune microenvironment may help us to both diagnose and treat more effectively. In our study, we showed that PLAU expression is positively correlated with stromal score. The stromal-immune score represents a prognosis stratification tool intended to be developed as reliable prognostic signatures in gastric cancer (*Wang, Wu & Chen, 2019a*). Aberrance of macrophage function significantly contributes to disease progressions, such as in the case of cancer, fibrosis, and diabetes (*Ngambenjawong, Gustafson & Pun, 2017*). By analyzing the immune cell proportions, we identified PLAU expression was positively correlated with M1 type macrophages, and negative association with CD4+ T cell, Tregs cell, and follicular helper T cell. These associations could explain the role of PLAU in HNSCC prognosis from the immunological respective. In addition, we found that PLAU expression was reduced in HPV positive HNSCC tumors as compared to HPV negative ones. HPV positivity is well-accepted as a strong survival favorable factor in head and neck cancer patients, which indirectly supports that low expression of PLAU predicts a better survival in HNSCC patients. And PLAU activity could be a partial reason for HPV's role in HNSCC tumors.

Back to clinical significance, the performances of PLAU and other independent prognosis indicators in predicting HNSCC 5-year survival outcome were investigated. Although HPV status, tumor size or neck node status is independent prognosis indicator in HNSCC, their AUC is very low according to the ROC assays. However, the AUC of PLAU expression reach 0.795, higher than other clinical parameters. Moreover, the combination of PLAU and neck node status could predict HNSCC 5-year overall survival outcomes with an 88.5% sensitivity and 64% specificity, which demonstrated the capability of PLAU expression in HNSCC prognosis.

Certainly, we need to be aware that our findings require the further validations from in vivo and vitro experiments although the conclusion was confirmed across five independent cohorts. Another limitation in this study is the platform used in different cohorts is different, which may produce bias to the data analysis and bring hardness for the deep integrated analysis.

## CONCLUSIONS

All in all, PLAU might be an independent biomarker for predicting outcomes of HNSCC patients. The elevated expression of PLAU was associated with HPV positivity and neck node status. PI3K-Akt pathway and aberrant proportions of immune cells might underly the mechanism of PLAU's oncogene role in HNSCC.

### Funding

This research was supported by the Project of Hunan Health Commission (B2019165), the National Natural Science Foundation of China (Nos. 81974424, 81874133, 81772903, and 81602389), the Natural Science Foundation of Hunan Province (Nos. 2020JJ4827, 2019JJ50944, and 2018JJ2630) and the Huxiang Young Talent Project (No. 2018RS3024). The funders had no role in study design, data collection and analysis, decision to publish, or preparation of the manuscript.

### Grant Disclosures

The following grant information was disclosed by the authors:
Project of Hunan Health Commission: B2019165.
National Natural Science Foundation of China: 81974424, 81874133, 81772903, 81602389.
Natural Science Foundation of Hunan Province: 2020JJ4827, 2019JJ50944, 2018JJ2630.
Huxiang Young Talent Project: 2018RS3024.

### Competing Interests

The authors declare there are no competing interests.

### Author Contributions

- Zhexuan Li performed the experiments, analyzed the data, prepared figures and/or tables, authored or reviewed drafts of the paper, and approved the final draft.
- Changhan Chen, Juncheng Wang, Guancheng Liu and Yuexiang Qin performed the experiments, prepared figures and/or tables, and approved the final draft.
- Ming Wei and Li She analyzed the data, prepared figures and/or tables, and approved the final draft.
- Yong Liu, Yongquan Tian, Gangcai Zhu and Xin Zhang conceived and designed the experiments, authored or reviewed drafts of the paper, and approved the final draft.
- Donghai Huang analyzed the data, authored or reviewed drafts of the paper, and approved the final draft.

### Data Availability

  Raw data and code are available in the Supplemental Materials.

### Supplemental Information

Supplemental information for this article can be found online at http://dx.doi.org/10.7717/peerj.10746#supplemental-information.

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
