# Peer review of "Overexpressed PLAU and its potential prognostic value in head and neck squamous cell carcinoma"

_PeerJ, doi:10.7717/peerj.10746_

## Round 0.1 · original submission · Major Revisions

Please revise the manuscript according to the reviewers' comments. In particular, the authors should better validate the expression of PLAU in their own specimens.

·

Basic reporting

Line 176. "higher expression of PLAU methylation" is not appropriate. It is not generally used "high (low) expression" for the status of DNA methylation status, but it is used in mRNA level.

Line 249. The authors mention PLAU expression is associated with radio-sensitivity. There is no evidence demonstrated in the study. They should correct the explanation.

Line 251. The limit of this analysis is that it has not been verified by in vitro/in vivo analysis using cell lines and animals, and is a retrospective analysis of the multiple datasets. In addition, there is no mention of the infection status of HPV (alternatively, p16 IHC), which is an important factor in head and neck cancer. Not only the lack of in vitro/in vivo analysis as the limitation of this study, the authors should also mention all the limitation points above in the discussion part.

Experimental design

Figure 1.
HPV (p16) status is an important prognostic factor in H&N cancer. It would be more informative if they could separate the tumor group by HPV (p16 IHC) status.

Figure 2.
E. It would be more informative if they could incorporate the HPV (p16 IHC) status into the multivariate analysis.
F. What was the unit of "expression of PLAU"? The terms "T-without" and "T-with" are vague. More detail clarification is needed.

Figure 3.
B. What does the color code (red, orange, yellow) mean?

Figure 4.
A&B. HPV (p16) status is an important prognostic factor in H&N cancer. It would be more informative if they could separate the tumor group by HPV (p16 IHC) status.
C. What was rationale for the cut-off of PLAU methylation status (high vs. low)?

In addition,it would be more informative if they could assess the correlation between mRNA expression and methylation value of PLAU gene.

Figure 5
It would be more informative if they could assess the correlation between PLAU expression and immune checkpoint molecules (PD-L1, PD-L2, CTLA4, LAG3, TIGIT etc.) in terms of immune related tumor microenvironment.

Table 1.
It is encouraged to incorporate HPV (p16 IHC) status in this clinical parameters as an important clinical factor.

Validity of the findings

Line 256. Conclusions
They mention that "PLAU can be used as a potential diagnosis and prognosis biomarker". However, based on the Cox multivariate analysis for OS [Figure 2E], the increased risk of high PLAU expression is only 18% despite it is significant, while the T-stage and N-stage are more prominent and convenient prognostic factors (52% and 74% increased risk, respectively with substantial significance). How can they emphasize the importance of PLAU as a prognostic biomarker from clinical perspective?
Based on the shown result, PLAU expression should not be used as a prognostic marker for the patient with HNSCC. The authors should reconsider the contents of conclusion.

Additional comments

The authors focused on the expression of the PLAU gene, one of the ECM pathways, and used datasets of multiple head and neck cancer cohorts. The association between PLAU expression in cancerous part, its effect on prognosis, estimation of microenvironment, and methylation status was shown based on data analysis.

Since different analysis tools are employed, it is understandable that the format of the output KM plot is different, but if possible a unified format would be easier for PeerJ readers.

Reviewer 2 ·

Basic reporting

1. The article conducted a comprehensive analysis of PLAU from the perspectives of expression, signaling pathways, methylation, tumor immune microenvironment, survival data, etc., but the content lacks necessary relation. The article describes the GEO database used for the expression level of PLUA, and the methylation level of PLUA uses the UALCAN and DiseaseMeth version 2.0 databases. Is the expression level of PLUA different between two databases?
2. There are clerical errors in references such as: Wu G et al. 2020.

Experimental design

1. The article hopes to prove that PULA is a potential biomarker for the diagnosis and prognosis of HNSCC, but the sensitivity, specificity, positive predictive value, negative predictive value, cut-off value and other indicators of PULA in HNSCC are not discussed, and clinical application is not enough.

Validity of the findings

1. This article uses multiple databases (TCGA, GEO, GTEx, STRING, DiseaseMeth version 2.0) and data analysis platforms (UALCAN, GEPIA, Kaplan-Meier survival analysis), data analysis methods (DiseaseMeth, Cytoscape, MethSurv, KEGG, GO, Gene Set Enrichment Analysis), the data source used in the article is more complex, the article does not mention the method of data quality control, how to ensure the reliability of the article results? The lack of consistency in the data used in each part of the article may introduce excessive bias and lead to unreliable results. It is recommended to consider this issue again.

Additional comments

In article, plasminogen activator urokinase (PLAU) is an important factor in extracellular matrix remodeling and metastasis. The article used a variety of data analysis tools to conduct a large number of analyses on multiple databases and obtained statistically different results. The related clinical and experimental data were described and analyzed retrospectively. However, the article has some problems as follows.
1. This article uses multiple databases (TCGA, GEO, GTEx, STRING, DiseaseMeth version 2.0) and data analysis platforms (UALCAN, GEPIA, Kaplan-Meier survival analysis), data analysis methods (DiseaseMeth, Cytoscape, MethSurv, KEGG, GO, Gene Set Enrichment Analysis), the data source used in the article is more complex, the article does not mention the method of data quality control, how to ensure the reliability of the article results? The lack of consistency in the data used in each part of the article may introduce excessive bias and lead to unreliable results. It is recommended to consider this issue again.
2. The article conducted a comprehensive analysis of PLAU from the perspectives of expression, signaling pathways, methylation, tumor immune microenvironment, survival data, etc., but the content lacks necessary relation. The article describes the GEO database used for the expression level of PLUA, and the methylation level of PLUA uses the UALCAN and DiseaseMeth version 2.0 databases. Is the expression level of PLUA different between two databases?
3. The article hopes to prove that PULA is a potential biomarker for the diagnosis and prognosis of HNSCC, but the sensitivity, specificity, positive predictive value, negative predictive value, cut-off value and other indicators of PULA in HNSCC are not discussed, and clinical application is not enough.
4. What is the basis for PLAU divided into high expression and low expression groups in table1? What is the cut-off value? Are the sample detection methods consistent in the database?
5. Among the clinical parameters of Table1, only metastasis of lymph nodes is statistically different, suggesting that PLAU may be related to lymph node metastasis. There is no significant statistical difference between tumor T staging and distant metastasis. How to understand the role of PLAU in the diagnosis and prognosis of HNSCC?
6. The article has a little problem with the statistical processing of data. In Table 1, the tumor staging and cervical lymph node metastasis, some cases are less than 5, which should be tested with continuous-corrected chi-square or Fisher's exact probability.
7. There are clerical errors in references such as: Wu G et al. 2020.

Reviewer 3 ·

Basic reporting

An important concern is that there are a lot of grammatical errors and typos in this manuscript. For examples:
- Using the clinical data from The Cancer Genome Atlas (TCGA) databases to explore whether PLAU as a potential prognostic factor in HNSCC.
- The relationship between the expression of PLAU and immune microenvironment were analyzed using stromal score or immune score.
- Moreover, it indicated that PLAU as a potential prognostic factor in HNSCC through ...
- No correlations between of PLAU with gender (P=0.0763) and grade (P=0.9461).
- ...
Therefore, the authors should re-check and revise carefully. I suggest that it needs to be checked by an English editing service.

Literature review is weak. The authors should refer more related works on bioinformatics studies on HNSCC.

What is "reanalysis of the data gathering" in line 76?

Experimental design

- The authors should release source codes for replicating the results.
- National Center for Biotechnology Information (NCBI) has been used to collect data in previously published works in bioinformatics such as PMID: 31319963 and PMID: 31920706. Therefore, the authors should refer more works into this description.
- For this kind of problem, ROC curves and AUC analyses should be conducted.

Validity of the findings

- "Conclusion" section is short and did not contain a lot of information.
- The authors should validate the performance on unseen data.
- The authors should compare the performance with previous works on the same dataset.

Additional comments

No comment

---

## Round 0.2 · accepted · Accept

The comments have been well addressed.

Reviewer 3 ·

Basic reporting

No comment

Experimental design

No comment

Validity of the findings

No comment

Additional comments

My previous comments have been addressed satisfactorily.